# Intelligent Eye-Controlled Electric Wheelchair Based on Estimating Visual Intentions Using One-Dimensional Convolutional Neural Network and Long Short-Term Memory

**DOI:** 10.3390/s23084028

**Published:** 2023-04-16

**Authors:** Sho Higa, Koji Yamada, Shihoko Kamisato

**Affiliations:** 1Graduate School of Engineering and Science, University of the Ryukyus, Nishihara 903-0213, Japan; 2Department of Engineering, University of the Ryukyus, Nishihara 903-0213, Japan; koji@ie.u-ryukyu.ac.jp; 3Department of Information and Communication Systems Engineering, National Institute of Technology, Okinawa College, Nago 905-2171, Japan; kamisato@okinawa-ct.ac.jp

**Keywords:** visual intentions, eye tracking, human intention recognition, electric wheelchair, machine learning, one-dimensional convolutional neural network, long short-term memory

## Abstract

When an electric wheelchair is operated using gaze motion, eye movements such as checking the environment and observing objects are also incorrectly recognized as input operations. This phenomenon is called the “Midas touch problem”, and classifying visual intentions is extremely important. In this paper, we develop a deep learning model that estimates the user’s visual intention in real time and an electric wheelchair control system that combines intention estimation and the gaze dwell time method. The proposed model consists of a 1DCNN-LSTM that estimates visual intention from feature vectors of 10 variables, such as eye movement, head movement, and distance to the fixation point. The evaluation experiments classifying four types of visual intentions show that the proposed model has the highest accuracy compared to other models. In addition, the results of the driving experiments of the electric wheelchair implementing the proposed model show that the user’s efforts to operate the wheelchair are reduced and that the operability of the wheelchair is improved compared to the traditional method. From these results, we concluded that visual intentions could be more accurately estimated by learning time series patterns from eye and head movement data.

## 1. Introduction

People with severe physical disabilities face many difficulties in their daily lives, such as moving around and eating. Research has been conducted on the operation of electric wheelchairs to improve the quality of life of these people, and various user interfaces have been developed [1]. In the voice control methods, the user operates the electric wheelchair by uttering “stop”, “go forward”, “left”, and “right” [2,3]. Moreover, the EEG (electroencephalography)-based brain–computer interface (BCI) processes the user’s EEG signals and converts them into control commands to drive the wheelchair [4]. These interfaces are an alternative to joysticks for people with severe physical disabilities. However, voice input may result in undesired commands due to noise and laborious movement adjustments. In the case of BCI, the user must always concentrate on the control commands, which places a heavy burden on the user [5].

On the other hand, research has been conducted on applying eye-tracking devices to interfaces because eye movements can operate equipment hands-free. Furthermore, human eye behavior is less susceptible to paralysis and strongly correlates with visual intentions. Typical eye movements include fixation and saccades. Fixation is an eye movement in which the user gazes at an arbitrary area, and a saccade is a rapid eye movement [6]. In these studies [7,8,9,10], fixation was detected from the user’s eye movements using eye tracking, and the electric wheelchair was driven in the fixated direction. However, people also move their heads and gaze when checking their surroundings or when focusing on a specific object. Such unintentional fixations are also incorrectly recognized as operations in an eye-tracking interface. This problem is known as the Midas touch problem and requires the classification of visual intentions [11,12].

One method to solve the Midas touch problem is in distinguishing fixation by setting an extended dwell time for making choices and decisions [13]. This method is called the gaze dwell time method, which selects arbitrary directions or objects by intentionally fixating on them for a certain time. We have developed an electric wheelchair that can steer in the fixated direction using the gaze dwell time method. Although the gaze dwell time method makes it easy for the user to check the surroundings while the wheelchair is moving, it requires a certain amount of time to distinguish fixation, which causes a lag when turning left or right. Furthermore, the longer gaze dwell time required for fixation detection forces the user to expend extra effort to keep fixating [1]. Thus, applying the results of fixation detection to the control system of an electric wheelchair makes the operation more difficult and increases the burden on the user. Hence, there is a need to develop a method to solve these problems.

In estimating visual intentions, the ideal approach is to real-time estimate them from subtle differences in the user’s natural eye movements [14,15]. Reseacrhers have considered that intentions are indicated in eye movement time series data, and have used machine learning models to estimate subjects’ intentions from eye movement patterns. In particular, Subramanian et al. used not only eye movements but also external environmental information, such as the depth value of the gazing object and the object’s name, to estimate the visual intention toward high accuracy. Doushi et al. [16] used the temporal relationship between the eye and head pose to determine the attentional state of the subject. They concluded from their experimental results that the head tends to move before the eye movement during task-oriented visual attention, such as lane changes. Other studies on the dynamics of head and eye gazing behavior show that when stimuli are presented in the visual field, the head tends to move later than the eye movements [17,18]. Thus, visual intentions tend to relate to head and eye movements. We were inspired by the findings of these related studies that head movements should be considered in addition to eye movements, and that depth information to the fixation point is necessary as the external environment information to estimate the visual intentions during the accurate driving of the electric wheelchair.

In this paper, we aim to develop a data-driven model that estimates in real time the user’s “moving intentions” and develops an eye-controlled electric wheelchair based on intention estimation. The system estimates in real time the visual intentions during the electric wheelchair operation and performs operation assistance according to intentions, which are expected to improve operability. For example, the user can turn right or left immediately according to the user’s intention to do so, without having to keep looking in the direction of movement for a certain period at the corner.

The rest of this paper is outlined as follows. Section 2 is a review of the related work. Section 3 is the Methodology section, which describes the architecture of the electric wheelchair system and the machine learning algorithm. Section 4 summarizes the performance of the visual intentions estimation model and the electric wheelchair control system. Finally, the discussion is stated in Section 5.

## 2. Related Work

Many studies have been conducted on integrating eye-tracking interfaces into electric wheelchair control systems, including video-based and EOG (electrooculography)-based methods.

In EOG-based eye-tracking methods [19,20], electrodes were placed on the user’s forehead or the skin around the eyes, and the gaze direction was detected by measuring and signal processing the electric potential when the user moves the eyeballs. The EOG-based method was less affected by light changes than the video-based eye-tracking method, but it is difficult to detect oblique eye rotation using this method.

In video-based eye-tracking methods [7,8,9,21,22], cameras were used to capture the user’s eyes and to track the position of the pupil via image processing. The pupil detection method depended on the camera type. In the infrared camera method, infrared light was irradiated onto the eyeball. The infrared light reflected by the cornea (Purkinje image) served as a base point, and the gaze direction was determined from the position between the pupil and the Purkinje image. This method had high detection accuracy, but if the infrared irradiation to the eye was missed, the pupil was not tracked. The method using a visible light camera determined the position of the pupil using image recognition from eye images taken under natural light conditions. This method was inexpensive and easy to handle compared to using an infrared camera. However, it required illumination with a certain brightness, and detection was difficult when the eyeball image was whited out or blacked out. Moreover, eye-tracking methods using video images have been studied in two ways: the camera was placed at a distance from the user (non-invasive), and the camera was placed directly on the user’s head or face (invasive).

In studies using invasive eye trackers, the eye tracker was placed on the user’s head to detect gaze direction [21]. The pupil was extracted using image processing, and the gaze direction was calculated from the motion trajectory of the pupil center coordinates. The pupil range of motion was divided into five regions and labeled. These five labels defined commands to control the movement of the wheelchair (backward, left turn, stop, right turn, and forward). A similar study also set five regions in the range of motion of the pupil and implemented left turn, right turn, forward, stop, and toggle switch (ON/OFF) functions so that the electric wheelchair was driven according to these regions [22]. Research has been conducted to detect eye direction without tracking the pupil [8]. Eye movements were classified by inputting eye images into convolutional neural networks (CNNs), then the electric wheelchair was driven according to the classification results (right, left, forward, and eye closed). Invasive eye trackers constantly track the eyes from a fixed angle regardless of the face direction. However, the eye tracker is attached directly to the user’s head, which causes a significant physical burden.

Studies using non-invasive eye trackers used an eye tracker placed in front of the user to detect gaze direction [9]. The user’s face was detected using face recognition, and then the eye region was determined from the facial landmarks. Next, four types of eye movements (look up, look left, look right, and look middle) were estimated from the eye images using transfer learning with VGG-16. The electric wheelchair was driven according to the estimated results. Research has also been conducted to place a monitor connected to the eye tracker in front of the user [7]. The eye tracker captured the user’s gaze on a monitor. The monitor displayed a front-facing image captured by the camera, and a control panel consisted of options and operation commands for controlling the electric wheelchair. The user operated the electric wheelchair through fixation on the control panel. Non-invasive eye trackers are less physically burdensome because the camera is located away from the user, but tracking is not available if the eyes are not visible to the camera due to the face pose.

The methods described so far divided the eyeball’s range of motion into an arbitrary number of regions or classified the eye movements to apply to control the electric wheelchair, but these methods led to misselection due to involuntary eye movements. This problem is called the Midas touch problem. A standard solution to overcome the Midas touch problem is to use eye-tracking with other modalities such as gaze dwell time [13], eye gestures [23], blinking [21], and switch and touchpad [1]. For example, the visual intentions are determined by blinking multiple times or by pressing a switch as an additional input to decide whether the person is moving to the destination of the gaze or checking the surroundings. Such multimodal input enables the system to correctly understand the visual intentions. However, the user is forced to concentrate excessively on the operation when driving on a complex route that requires repeated checks and stops. In other cases, the user is severely disabled if they cannot move their upper limbs. Therefore, research has been conducted to infer visual intentions from natural gaze behavior.

Inoue et al. [14] developed a classification model for classifying cooking operations from eye movement patterns. They used N-grams to describe gaze patterns such as eye movements, fixations, and blinks during cooking, and trained and classified gaze patterns using a random forest. The mean accuracy rate of the trained random forest was 66.2% for all of the cooking motions. Other approaches have also been studied. Ishii et al. [24] proposed an algorithm to estimate the user’s level of engagement in a conversation based on gaze information such as eye movement patterns, fixation time, amount of eye movement, and pupil size. They trained a decision tree as an engagement estimation model and predicted user disengagement with about 70% accuracy. These studies tried to estimate visual intentions using a data-driven approach that trains the model using only features related to eye movements and succeeded in estimating them with moderate accuracy.

In recent years, RGB-D cameras and laser rangefinders (LRFs) at low cost and the advancement of object detection algorithms have made obtaining information on fixating targets easy. Hence, research on visual intentions estimation, considering the fixating object, has been conducted. Huang et al. [15] studied the prediction of which ingredients a customer would request based on a customer’s gaze cues, in a collaborative process in which the worker makes a sandwich using the ingredients requested by the customer. They collected and analyzed data from a simulated sandwich-making process, and found that the intention to select an ingredient was represented by features such as the number of fixations on the ingredient, the time per fixation, the total fixation time, and the most recently seen ingredient. These features were trained on the SVM classifier, which successfully predicted the customer’s order intention with 76% accuracy. Furthermore, the classifier made the correct prediction approximately 1.8 s before the customer’s voiced request.

A recent work by Subramanian et al. [25] indicated that visual intentions to objects were inferred from a wheelchair user’s natural eye movements during an electric wheelchair navigation task. Moreover, they successfully applied visual intentions to wheelchair steering control. First, they performed a task with and without interactive intentions toward objects on subjects and recorded their fixation points during the task. An object detection model based on single shot multibox detector (SSD) [26] and MobileNets [27] architecture was utilized to compute object labels and bounding boxes. Next, the SVM and weighted K-nearest neighbors (KNNs) were trained on the fixation points on the object. These classifiers output Boolean values for interactive or non-interactive intention, and the classification accuracy was generally higher than 78.8%. The visual intentions were estimated each time the user looked at an object, and the integrated system autonomously navigated the wheelchair to the object’s location.

Thus, information about the fixating object has become an essential element in estimating visual intentions. In addition, the remarkable advancement of deep learning-related technologies has led to the development of high-performance regression and classification algorithms for time series data. Therefore, we estimate visual intentions by training a deep learning model on patterns of the eye and head movement, fixation, and depth information to the fixation point.

## 3. Materials and Methods

### 3.1. The Electric Wheelchair Control System

Figure 1 shows the appearance of the eye-controlled electric wheelchair and the system components. We use the WHILL model CR, a research and development model electric wheelchair manufactured by WHILL Inc. The single-board computer uses a JetsonTX2 manufactured by NVIDIA. The tablet device used is an Apple iPad Pro, placed in front of the wheelchair user to capture facial images every 0.2 s. The measurement range of the camera is 3.14 rad horizontally and 2.09 rad vertically. The LRF is URG-04LX-UG01, manufactured by HOKUYO. The scanning angle is 4.18 rad, and the measurement range is 0.02 m to 5.6 m. The LRF obtains depth values to the points facing the user’s eyes and head. WHILL’s joystick module converts X-axis input values to angular velocity, and Y-axis input values to translational velocity. The electric wheelchair is controlled by bypassing this joystick module and instead inputting speed commands from the electric wheelchair control system implemented in JetsonTX2.

The control system comprises a visual intentions estimation model and a gaze dwell time method. The visual intentions estimation model is data-driven and can be customized for many tasks by adding more driving scenario data during the training phase. First, we construct a simple model by limiting the tasks to be estimated to “forward”, “right turn”, “left turn”, and “stop” only, and we verify whether intention estimation is possible. The speed command is calculated based on the rotation angle of the eye at the time of intention estimation. Moreover, the gaze dwell time method is applied when driving straight ahead to avoid unintended movement in the direction when looking aside or checking the surroundings.

Figure 2 shows the flow from the data acquisition of various sensors to driving.

1.The depth values to the points facing the user’s eyes and head are facing are acquired using LRF. The depth values are input to the visual intentions estimation model.2.The rotation angles of the horizontal and vertical axes of the user’s eyes and head are obtained from the camera image at each time t using ARKit Face Tracking, a library for measuring face and eye posture provided by Apple Inc. (Cupertino, CA, USA) [28]. The rotation angles are noise-eliminated using a five-point moving average filter. The rotation angles of the horizontal and vertical axes are treated as a set of vectors in a two-dimensional plane. Next, the angular velocity, standard deviation, and dwell time are calculated from the amount of change in these vectors. The rotation angle vector, angular velocity, standard deviation, and dwell time are input to the proposed model.3.The proposed model outputs the control commands “forward”, “turn left”, “turn right”, and “stop” at every time t according to the input values.4.The gaze dwell time method is applied when going straight ahead, and the wheelchair drives in the direction where the user has gazed for more than 0.7 s. If the dwell time is less than 0.7 s, the wheelchair drives at the previous speed command.5.The following equations calculate the speed command input to the joystick module of the electric wheelchair. $\theta_{y(t)}$ is the rotation angle of the eyeball on the vertical axis at the time that the command is output by the visual intentions estimation model, and $\theta_{x(t)}$ is the eye rotation angle of the horizontal axis. $\theta_{y(t)}$ is used as a switch to stop and start according to the threshold value th, as shown in Equation (Equation 1). We set th = 0.35 rad so that the electric wheelchair stops when the user looks at the bottom of the tablet device. *c* in Equation (Equation 1) is a constant, and in this paper, *c* is set to 100. The translational speed of the electric wheelchair is constant when the wheelchair is driven. In Equation (Equation 2), α is a coefficient for adjusting the angular velocity. However, the $X_{\left(t\right)}$ and $Y_{\left(t\right)}$ values are 0 only when the model outputs a stop command.
(1)$$Y_{\left(t\right)}=\left\{\begin{array}{cc} c & \theta_{y(t)}>th\\ 0 & \theta_{y(t)}\leq th \end{array}\right.$$
(2)$$X_{\left(t\right)}=\alpha \;\cdot \;\theta_{x(t)}$$6.The speed command values $X_{\left(t\right)}$ and $Y_{\left(t\right)}$ are input to WHILL, which converts $X_{\left(t\right)}$ to an angular velocity and $Y_{\left(t\right)}$ to a translational velocity to drive the motor.

The above process enables the electric wheelchair to drive following the eye’s direction. The movement direction of the electric wheelchair is limited to forward, right/left turn, and stop, only to ensure safety, because physically disabled persons cannot quickly check backwards due to muscle paralysis or rigidity.

### 3.2. Generation of Feature Vectors

This section describes the feature vectors input to the visual intentions estimation model. First, the rotation angles of the horizontal and vertical axes of the eyes and head are treated as vectors $E=(Eye_{x},Eye_{y})$ and $H=(Head_{x},Head_{y})$, respectively, in the two-dimensional plane. Next, the angular velocity and the standard deviation are calculated from the amount of change in these vectors. Moreover, the “attention histogram” proposed by Adachi et al. [13] is used to calculate the “gaze dwell time”, a feature that easily expresses the psychological state.

The attention histogram is a two-dimensional histogram that indicates the fixation intensity, and the histogram is distributed around the fixation point. As shown in Equation (Equation 3), a two-dimensional Gaussian distribution is used for the distribution. The two-dimensional Gaussian distribution is calculated from the angles $\phi_{m}$ and $\phi_{n}$ between the fixation point and the other points shown in Figure 3. The histogram is obtained by adding up this distribution every time. Equation (Equation 4) shows the formula for the attention histogram.
(3)$$g_{ij}^{\left(t\right)}=\frac{1}{2\pi \sigma^{2}}\;\cdot \;\exp\left(-\frac{{\phi_{m}}^{2}+{\phi_{n}}^{2}}{2\sigma^{2}}\right)$$
(4)$$w_{ij}^{\left(t\right)}=\alpha \;\cdot \;w_{ij}^{\left(t-1\right)}+g_{ij}^{\left(t\right)}$$
where $w_{ij}^{\left(t\right)}$ is the histogram frequency of point $P_{ij}$ at current time *t*, α is the attenuation rate.

When the user keeps fixating on the same point, the histogram frequency reaches $\frac{1-\alpha }{2\pi \sigma^{2}}$, which is the peak value. When the histogram frequency reaches 90% of the peak value, the user is judged to have fixated on the point, and the gaze dwell time starts counting. When the histogram frequency falls below 50% of the peak value, the gaze dwell time is reset. The attention histogram determines the time required to perform the fixate detection using the parameters σ and α. In this study, we set the standard deviation σ to 5 and the attenuation rate α to 0.5 to obtain a fixate detection time of 0.7 s. The attention histogram was also applied to the gaze dwell time method implemented in the electric wheelchair control system.

Finally, the system does not know where the user is gazing because the eye and head tracking data are vectors in a two-dimensional plane. For example, if the user looks at a wall when the electric wheelchair stops, the system does not determine whether that is an open or an obstructed space, based on the eye rotation angle vector alone. Environmental information, such as the distance to the object looked at, is extremely important to determine whether the user interacts intentionally. Therefore, we used each depth value to the point where the user’s eyes and head are facing as a feature value.

Thus, the feature vector input to the visual intentions estimation model are multivariate time series data with 10 dimensions: the rotation angle vector, angular velocity, standard deviation, dwell time, and depth values for each eye and the head.

### 3.3. One-Dimensional Convolution Neural Network

CNN (convolutional neural network) is one of the deep learning methods using human neural models. CNN extracts local features from the given data via the convolution and pooling layers and performs regression and classification based on the local feature values in a fully connected layer [29,30,31]. CNN is characterized by robustness to input a signal shift [32] and is widely applied for image processing, object detection, natural language processing, and biometric signal processing [33]. Since the eye and head movements are one-dimensional sequence data, we use a one-dimensional CNN (1DCNN) model in which the convolution and pooling layers have a one-dimensional shape. Detailed explanations of 1DCNN are described below. Figure 4 shows the architecture of a 1DCNN model.

#### 3.3.1. Convolution Layer

The convolution layer extracts local features from the local region of the input signal and the upper layer’s feature map, generating a feature map that summarizes the local features [34,35]. This mechanism is known as local receptive fields. Figure 4 shows the architecture of the 1DCNN model. The feature map is output by multiplying the input layers with the convolution kernels and inputting their summed values to the activation function. In this process, a convolution kernel with the same weights is used for all convolutions in each local region. It is known as weight sharing. Local receptive fields and weight sharing effectively reduce the parameters, such as the convolution kernels’ weights and biases, thereby reducing the complexity of the network structure. Generally, convolutional processing uses multiple convolution kernels to extract the various features that contribute to classification. The one-dimensional convolution process is calculated as in Equation (Equation 5).
(5)$$x_{j}^{l}=f\left(\sum_{c}\sum_{i=1}^{M^{l-1}}x_{i}^{\left(l-1,c\right)}*w_{ij}^{\left(l,c\right)}+b_{j}^{l}\right)$$
where $x_{j}^{l}$ is the value of the *j*th neuron in layer *l*, and $b_{j}^{l}$ is the bias corresponding to the convolution kernel. $w_{ij}^{\left(l,c\right)}$ is the *i*th kernel weight linked to the *j*th neuron in layer *l* of the input channel *c*, and $x_{i}^{\left(l-1,c\right)}$ denotes the input value from the *i*th neuron in layer l−1 to the *j*th neuron in layer *l* of the input channel *c*. The kernel weights $w_{ij}^{\left(l,c\right)}$ and bias $b_{j}^{l}$ are obtained using backpropagation [36]. *c* is the number of input channels, $M^{l-1}$ indicates the number of neurons within the selected range of the feature map in layer l−1, ∗ represents the convolution operator, and f() denotes the activation function. In our paper, ReLU [30] is selected as the activation function. ReLU is described in Equation (Equation 6).
(6)$$f\left(x\right)=\left\{\begin{array}{cc} 0 & x\leq 0\\ x & x>0 \end{array}\right.$$

#### 3.3.2. Pooling Layer

The pooling layer subsamples the features extracted from the upper convolution layer and aggregates the local features. We use the max pooling [37], which outputs the largest parameter in the feature map within a window range as a feature. Figure 4 shows how the max pooling method reduces the feature map obtained in the previous layer by half. The pooling layer reduces the dimension of the features and thus avoids overfitting [38]. Furthermore, the pooling layer can retain important features while reducing noise. Hence, a CNN has the robustness to input signal shifts (translational invariance). Max pooling is defined as follows.
(7)$$a_{j}^{l}=\max_{i\in R_{j}^{l}}x_{i}^{l}$$
where $R_{j}^{l}$ represents the index of the *j*th pooling region in the *l*th layer, and $a_{j}^{l}$ is the output feature of the *j*th pooling region in the *l*th layer. $x_{i}^{l}$ denotes the input features of index *i* in the *l*th layer.

#### 3.3.3. Fully Connected Layer

Feature maps extracted through multiple convolutions and pooling layers are given to the fully connected layer [39,40], whereas the convolution layer is a sparse network architecture; all of the neurons in the fully connected layer are interconnected with the neurons in the previous layer and the neurons in the next layer [41]. The output value of each neuron is calculated as in Equation (Equation 8). The output values are provided to the next layer.
(8)$$z_{j}=f\left(\sum_{i=1}^{M}x_{i}\;\cdot \;w_{ij}+b_{j}\right)$$
where $z_{j}$ is the output value of the *j*th neuron, and $b_{j}$ is the bias value at the *j*th neuron. $w_{ij}$ represents the *i*th weight value linked to the *j*th neuron. $x_{i}$ denotes the input value from the *i*th neuron, and *M* is the number of neurons. f() denotes the activation function.

#### 3.3.4. Output Layer

A softmax function is adopted for the output layer. The softmax function is described in Equation (Equation 9).
(9)$$p_{i}=\frac{e^{z_{i}}}{\sum_{k=1}^{M}e^{z_{k}}}$$
where $p_{i}$ is the value of the *i*th output unit, $z_{k}$ is the value of the *k*th neuron, and *M* is the number of neurons. The number of neurons equals the number of predicted labels. The value of $p_{i}$ indicates the probability because the sum of each $p_{i}$ in the output layer is equal to 1. Finally, the index of the unit with the highest value is output as the predictive label $\hat{y}$, as shown in Equation (Equation 10).
(10)$$\hat{y}=\underset{i}{arg max}p_{i}$$

### 3.4. Long Short-Term Memory

The RNN (recurrent neural network) is a neural network model with a recurrent structure in the middle layer. The RNN can process data considering past states because the previous output is used as an input value. Hence, RNN is used to forecast time series data [42]. In particular, LSTM (long short-term memory) is a type of RNN with a unit called LSTM block in the middle layer that learns contexts with long-term dependencies from sequences [43,44]. This model is widely utilized in natural language processing [45] and speech recognition [46]. Figure 5 shows a schematic diagram of an LSTM block.

LSTM consists of a unit for storing information, called the cell state, and three gate mechanisms (forget gates f(t), input gates i(t), and output gates o(t)) that control the information. The cell state is an essential component for LSTMs, which store and pass information between LSTMs by retaining or erasing old information in the cell state and adding new information. The forget gates f(t) control the amount of old information erased. The forget gates f(t) are shown in the equation below.
(11)$$f_{\left(t\right)}=\sigma \left(W_{f}\left[h_{\left(t-1\right)},x_{\left(t\right)}\right]+b_{f}\right)$$
where $h_{\left(t-1\right)}$ is the output value of the middle layer at the previous time, and $x_{\left(t\right)}$ is the input value at the current time *t*. σ denotes the sigmoid function. $f_{\left(t\right)}$ is multiplied by the old cell state $c_{\left(t-1\right)}$ to determine how much information to erase.

The new information added to the cell state is selected through the following process. First, the candidate values ${\tilde{c}}_{\left(t\right)}$ of information added to the cell state $c_{\left(t\right)}$ are calculated as shown in the following equation.
(12)$${\tilde{c}}_{\left(t\right)}=tanh\left(W_{c}\left[h_{\left(t-1\right)},x_{\left(t\right)}\right]+b_{c}\right)$$
where tanh represents the hyperbolic tangent function.

Next, the input gates $i_{\left(t\right)}$ determine the amount of candidate value ${\tilde{c}}_{\left(t\right)}$ to be added to the cell state $c_{\left(t\right)}$. The input gates $i_{\left(t\right)}$ are defined as follows.
(13)$$i_{\left(t\right)}=\sigma \left(W_{i}\left[h_{\left(t-1\right)},x_{\left(t\right)}\right]+b_{i}\right)$$

The cell state $c_{\left(t\right)}$ is updated using the obtained $f_{\left(t\right)}$, $i_{\left(t\right)}$, ${\tilde{c}}_{\left(t\right)}$, $c_{\left(t-1\right)}$. The updated equation is shown below.
(14)$$c_{\left(t\right)}=f_{\left(t\right)}\;\cdot \;c_{\left(t-1\right)}+i_{\left(t\right)}\;\cdot \;{\tilde{c}}_{\left(t\right)}$$

Finally, the output value $h_{\left(t\right)}$ of the LSTM block is obtained by passing the updated cell state value $c_{\left(t\right)}$ through the output gates $o_{\left(t\right)}$. The equations for the output gates $o_{\left(t\right)}$ and the output value $h_{\left(t\right)}$ are shown below.
(15)$$o_{\left(t\right)}=\sigma \left(W_{o}\left[h_{\left(t-1\right)},x_{\left(t\right)}\right]+b_{o}\right)$$
(16)$$h_{\left(t\right)}=o_{\left(t\right)}\;\cdot \;tanh\left(c_{\left(t\right)}\right)$$

The updated cell state $c_{\left(t\right)}$ and the newly generated output value $h_{\left(t\right)}$ are passed to the next time step. In the previous equations, $W_{f}$, $W_{c}$, $W_{i}$, and $W_{o}$ represent weights, and $b_{f}$, $b_{c}$, $b_{i}$, and $b_{o}$ represent biases.

### 3.5. Proposed Method for Estimating Visual Intentions Using One-Dimensional Convolutional Neural Network and Long Short-Term Memory

This section describes the 1DCNN-LSTM (one-dimensional convolutional neural network and long short-term memory). 1DCNN has strengths in local feature extraction, while RNN has advantages in mining time series data. Therefore, we focus on the advantages of both 1DCNN and LSTM and combine the two models to develop a 1DCNN-LSTM model. Table 1 shows the main structural differences between 1DCNN, LSTM, and 1DCNN-LSTM. 1DCNN excels at extracting local features through convolutional and pooling layers, and LSTM excels at processing sequence data through recurrent neural networks. 1DCNN-LSTM combines these two model structures, preserving their respective characteristics. In the 1DCNN-LSTM model, the 1DCNN is tasked with extracting useful local features from the feature vector in the first stage, and the LSTM is tasked with learning the long-term dependencies of the feature sequences in the next stage so that the model can infer based on local features, taking into account time series information. The 1DCNN-LSTM extracts and learns local spatial and temporal features from multivariate time series data, and many promising results have been reported [47,48,49,50,51]. Thus, 1DCNN-LSTM is expected to achieve a high degree of classification accuracy for human intention recognition in this study.

The network structure of the constructed 1DCNN-LSTM model is shown in Figure 6. The 1DCNN-LSTM model consists of an input layer, one CNN layer, two LSTM layers, two fully connected layers, and an output layer. First, the 1DCNN-LSTM model is input with multivariate time series data of 10 variables. Next, 1DCNN extracts the local features from the input data and provides them to the LSTM. One-dimensional CNN downsamples the data length at the feature extraction from the data; the time series data is reduced to half its length. The 1DCNN layer consists of one convolution layer and one pooling layer. The number of convolution kernels is 48, the kernel window width is 2, the stride width is 1, the pooling width is 2, and the activation function used is ReLU.

Next, the LSTM layer extracts temporal features from the unique feature map provided by the 1DCNN layer. In this LSTM layer, each LSTM block extracts features at each time step, resulting in a two-dimensional (time step × number of LSTM blocks) feature map. These feature maps are converted to one dimension through the flattened layer, and are fed to the fully coupled layer. The LSTM layers are stacked; each layer has 64 LSTM blocks, and the activation functions used are hyperbolic tangent and sigmoid functions.

Two fully connected layers are connected next to the LSTM layer. The first fully connected layer has 48 neurons and the second fully connected layer has 16 neurons, and ReLU is applied as the activation function. Dropout [52] is utilized to the fully connected layer to avoid overfitting. Dropout is a method to improve the model’s generalization performance by randomly invalidating some neurons from the network according to a certain amount of probability during the learning process. Invalidated neurons have an output value of zero or no output value [53,54]. The white neurons in the fully coupled layer in Figure 6 indicate the invalidated neurons. The dropout rate is set to 0.05.

Finally, the output layer is placed at the end of the model. The output layer has four neurons, to which a softmax function is applied to calculate the probability for each label (forward, right turn, left turn, and stop). The hyperparameters for each layer were acquired using hyperparameter optimization, as described in Section 4.1.2.

The model’s parameters (weights and biases) that minimize the error in the output values are acquired via the supervised learning of feature vectors’ patterns corresponding to the visual intentions. After learning, when the feature vectors are input to the trained model, the output values are determined corresponding to the input patterns using the acquired parameters. The learning process is described below. First, parameters are initialized to random values, and the input values are propagated through the input layer to the output layer to obtain the output values. Next, the gradient for each parameter is calculated from the error in the output values using the error backpropagation method [55]. Finally, new parameters are obtained and updated based on the gradients and the optimization algorithm. This process is repeated until the tolerance value is satisfied or the maximum number of iterations is reached. Thus, optimizing the parameters minimizes the error and improves the prediction accuracy. We use the cross-entropy loss function to calculate the error, and Adam [56] is used as the optimization algorithm. The cross-entropy loss function is represented by Equation (Equation 17), and Adam is defined following Equations (18)–(22).
(17)$$E=-\sum_{i=1}^{N}t_{i}\log y_{i}$$
where $y_{i}$ represents the predicted values, $t_{i}$ denotes the ground truth, and *N* is the total number of sample data.
(18)$$m_{t}=\beta_{1}m_{t-1}+\left(1-\beta_{1}\right)\frac{\partial E}{\partial w}$$
(19)$$v_{t}=\beta_{2}v_{t-1}+\left(1-\beta_{2}\right){\left(\frac{\partial E}{\partial w}\right)}^{2}$$
(20)$$\hat{m_{t}}=\frac{m_{t}}{1-\beta_{1}^{t}}$$
(21)$$\hat{v_{t}}=\frac{v_{t}}{1-\beta_{2}^{t}}$$
(22)$$w\leftarrow w-\eta \frac{\hat{m_{t}}}{\sqrt{\hat{v_{t}}}+\epsilon }$$

A moving average $m_{t}$ of the gradient is obtained by Equation (Equation 18), where $m_{t-1}$ is the previous gradient, *E* is the cross-entropy loss, and *w* is the parameter. Similarly, Equation (Equation 19) shows a moving average $v_{t}$ of the squared gradient, where $\beta_{1}$ and $\beta_{2}$ denote attenuation rates, and the initial values of $m_{t}$ and $v_{t}$ are 0. As $m_{t}$ and $v_{t}$ are minute values at the early stage of updating, the values are adjusted according to Equations (20) and (21). In Equations (20) and (21), *t* represents the number of updates, and as the updates proceed, $\beta_{1}^{t}$ and $\beta_{2}^{t}$ asymptotically approach 0, and so the effect of this calculation becomes minimal. Finally, the parameter *w* is updated using Equation (Equation 22), where η is the learning rate and ϵ is a small value to avoid zero division. When the gradient changes significantly, the learning rate η is decreased by $\hat{m_{t}}$ and $\hat{v_{t}}$ to suppress significant changes in the parameter *w*, thus making the parameter update more efficient. In this paper, we set the parameters $\beta_{1}=0.9$, $\beta_{2}=0.999$, η=0.001, and $\epsilon =10^{-8}$ according to the recommended values, using reference [56].

### 3.6. Overall of the Visual Intention Estimation Method

A flowchart of the visual intention estimation method based on machine learning models proposed in this paper is shown in Figure 7. First, the signals measured by each sensor are normalized to construct a data set with a set of predictive labels. This data set is split into a training set, a validation set and a test set in a certain proportion. The model is then built according to the pre-defined hyperparameters (convolutional layer, LSTM layer, fully connected layers, number of filters, kernel size, and activation function) and the weight parameters *w*, *b* are initialized. Next, the model training is started using the training data and the validation data model. At this point, time series cross-validation (K=5) is used to evaluate the generalizability of the model. Finally, the performance of the models is evaluated by inputting the test data set into the *K* trained models and calculating the F1-score from the difference between the output values and the ground truth. The models to be built are 1DCNN-LSTM, 1DCNN, and LSTM, and their performance is compared.

## 4. Experiments

Two experiments are conducted: one is a comparative experimental evaluation with two other existing models, i.e., LSTM and 1DCNN, to demonstrate the effectiveness of the proposed method. A second experiment evaluates an electric wheelchair control system’s operability.

### 4.1. Evaluation of the Effectiveness of the 1DCNN-LSTM

#### 4.1.1. Data Sets and Pre-Processing

The data set is used for the supervised learning and validation of the model and consists of objective and explanatory variables. The explanatory variables are the feature vectors described in Section 3.2. The objective variable is the class label *L* assigned according to the joystick’s horizontal angle $\theta_{x}$ and vertical angle $\theta_{y}$. The class label *L* is determined according to Equation (Equation 23).
(23)$$L=\left\{\begin{array}{cc} Left & 0\;\mathrm{rad}\leq \theta_{x}<1.396\;\mathrm{rad}\\ Forward & 1.396\;\mathrm{rad}\leq \theta_{x}\leq 1.745\;\mathrm{rad}\\ Right & 1.745\;\mathrm{rad}<\theta_{x}\leq 3.142\;\mathrm{rad}\\ Stop & \theta_{x}=0\;\mathrm{rad}\wedge \;\theta_{y}=0\;\mathrm{rad} \end{array}\right.$$

Feature vectors such as the rotation angle vector, angular velocity, standard deviation, dwell time, and depth value have different units and scales. Since different scales among feature vectors significantly affect model training, the whole data set is normalized so that the minimum value is 0 and the maximum value is 1. We asked 10 subjects aged 20–29 years old to ride in an electric wheelchair on a course that included turning right and left, moving forward, and stopping to collect eye and head movements data for creating training data sets (sample size: 45,000). Similarly, we asked the subjects to ride on a course different from the training data set to create test data sets (number of samples: 4527) for model evaluation.

#### 4.1.2. Hyperparameter Optimization

Hyperparameters mean parameters such as the number of hidden layers in the neural network, and the number of neurons in each network layer. These values affect the prediction accuracy of the model. Unlike weights and biases, which are acquired through learning, the hyperparameters strongly depend on the researchers’ experience and subjectivity in determining values. Hyperparameter optimization methods include grid search, random search, and Bayesian optimization. These algorithms select hyperparameter values from a specified range and use those values to perform the learning process. This learning process is repeated multiple times, and the hyperparameter value that minimizes the error between the output value and the ground truth is finally selected. Hyperparameter optimization is also typically performed in conjunction with cross-validation. Cross-validation avoids that the hyperparameters are optimized only for specific data, thus obtaining hyperparameters that can be estimated with a high degree of accuracy, even for unknown data. Cross-validation is described in Section 4.1.3. Because the model learning process is very time-consuming and our computational resources are limited, we use Bayesian optimization for the hyperparameter optimization method. Better results with fewer experiments have been reported using Bayesian optimization [57,58,59]. In this paper, we utilize Hyperopt, a library in Python [59].

#### 4.1.3. Evaluation Metrics and Methodology

Accuracy, a measure of what percentage of the total data set could be classified correctly, is often used to evaluate classifier performance. However, if the distribution of labels in the data sets is unbalanced, accuracy cannot be appropriately measured because labels with only a tiny amount have little effect on accuracy compared to vast amounts of labels [60,61]. The data sets collected are unbalanced, with many “Forward” labels compared to the other labels. If the proposed model only outputs “Forward” for all data, a high value of accuracy would be obtained, but it would be an inadequate evaluation because the model does not output the other labels. Hence, the F1-score is employed to evaluate the model performance. F1-score is the harmonic mean of the metrics called Recall and Precision, where a value that is closer to 1 means a better model performance and less misclassification for each label [62]. The F1-score is calculated via the following equations.
(24)$$Recall=\frac{TP}{TP+FN}$$
(25)$$Precision=\frac{TP}{TP+FP}$$
(26)$$F1-score=\frac{2\;\cdot \;Recall\;\cdot \;Precision}{Recall+Precision}$$
where TP denotes true positives, TN denotes true negatives, FP represents false positives, and FN represents false negatives.

In addition, several experimental conditions are also established to gain a deeper understanding of the effectiveness of the 1DCNN-LSTM model. The first condition is the time series data length (time step). We test how the performance of the model varies with the length of the time series data given to the model. Thus, the input time series data lengths are set to 0.5 s, 1 s, 2 s, 3 s, and 4 s. The second condition is the type of data set. First, a data set that combines eye and head movements, a data set that contains only eye movements, and a data set that contains only head movements are prepared. Then, we train 1DCNN-LSTM, LSTM, and 1DCNN with a combination of each data set type and time series data length and analyze model performance.

We set individual hyperparameters for all models and train the models using Tensorflow and Keras. The learning rate is 0.001, and the batch size is 50. Model training and performance evaluation are performed with k-fold time-split cross-validation [63]. The flow of k-fold time-split cross-validation is shown below.

1.The data set is divided into k in time series order. The kth data set contains the k-1th data set. The first data set has the least amount of data, and the last has the same amount as the original data set before the division. In this work, the number of splits k is set to 5. Thus, the data set with k0 has 25,000 samples; similarly, k1 has 30,000, k2 has 35,000, k3 has 40,000, and k4 has 45,000 samples.2.Each data set is divided into validation data (number of samples: 5000) and training data (number of total data−number of validation data) to train the model. The training data are divided into past time series, and the validation data into future time series.3.Repeat model training k times using the divided data set to construct k-trained models.4.Input test data (number of samples: 4527) for model evaluation into k-trained models, and calculate the F1-score from the classification results.5.The F1-score are averaged to obtain the final performance evaluation.

Finally, permutation feature importance (PFI) [64,65,66] is performed on the model with the highest F1-score to show which features contribute to the visual intentions estimation. First, the test datasets are input to the trained model, and the cross-entropy error between the output value and the ground truth is calculated and used as a baseline. Next, one feature is selected from the data set and is randomly permuted. Subsequently, the permuted data set is input to the trained model, and the cross-entropy error is calculated. If the error value is larger than the baseline, the model depends on those features. The larger the error value indicates, the more influential the feature. This mechanism is applied to all features to calculate feature importance.

#### 4.1.4. Results

Table 2 shows the comparative results of the mean F1-score of each model. The “head + eye” listed in the Models (Data sets) column of the table shows that the explanatory variables are 10 variables (the rotation angle vector, angular velocity, standard deviation, dwell time, and depth values for each eye and the head). Similarly, the “head” data set consists of 5 variables (the rotation angle vector, angular velocity, standard deviation, dwell time, and depth values for the head) and the “eye” data set consists of 5 variables (the rotation The “eye” is a data set consisting of five variables (the rotation angle vector, angular velocity, standard deviation, dwell time, and depth values for the eye).

The highest evaluated value of each model shows that the F1-score of LSTM (time step = 3 s, data sets = head + eye) is 0.912, the F1-score of 1DCNN (time step = 3 s, data sets = head + eye) is 0.915, and 1DCNN-LSTM (time step = 3 s, data sets = head + eye) has an F1-score of 0.918, with 1DCNN-LSTM having the best evaluation value. The 1DCNN-LSTM model using the data set combining eye and head movements performed better than 1DCNN-LSTM using the other data sets, except at 0.5 s. The same trend was observed in the evaluation results of 1DCNN when the length of the input time series was between 0.5 s and 3 s, and in the evaluation results of LSTM for all of the input time series. The F1-score also improved as the input time series increased, and the F1-score of each model reached a peak value within 2 s to 4 s.

Table 3 shows the Precision, Recall, and F1-score associated with the prediction labels for each fold that led to the calculation of the mean F1-score for the 1DCNN-LSTM time step = 3 s, data sets = head + eye). For all folds, the F1-score of “Forward” and “Stop” was higher than 0.9, while “Left” and “Right” had lower values than “Forward” and “Stop”. The mean of F1-score of all folds was above 0.91, and there were no extreme differences between folds, indicating that the visual intention estimation model has excellent generalizability.

Figure 8 and Figure 9 show the visual intentions estimation graphs for the test data 0 s to 200 s and 200 s to 400 s intervals. In comparing each figure, we observe that the 1DCNN-LSTM model estimates more accurately than the other models and has fewer misclassifications. Table A1, Table A2 and Table A3 show the hyperparameters for 1DCNN-LSTM (time step = 3 s, data sets = head + eye), 1DCNN (time step = 3 s, data sets = head + eye), and LSTM (time step = 3 s, data sets = head + eye).

Next, Figure 10 shows the results of the permutation feature importance. The most important feature that contributes to the estimation of visual intentions is “Rotation angle vector (Eye)”, with a value of 0.391. On the other hand, the “Dwell time(Head)” has the lowest value of 0.173. The feature importances of “Depth value(Eye)” and “Depth value(Head)” were 0.359 and 0.324, ranking second and third among all of the features. These results indicate that environmental information contributes significantly to visual intention estimation in addition to eye movements.

### 4.2. Evaluation of Electric Wheelchair Operability

#### 4.2.1. Evaluation Metrics and Methodology

The 1DCNN-LSTM model (time step = 3 s, data sets = head + eye) with the highest accuracy in the evaluation experiments described in Section 4.1 is implemented in an electric wheelchair control system, and the operability is evaluated through driving experiments. The comparison method is the gaze dwell time method, where the time required to detect fixation is 0.7 s. Fourteen subjects aged 20–29 years are asked to drive the course shown in Figure 11 using the respective electric wheelchair control systems. For the first drive, the electric wheelchair is operated using the traditional method, and the proposed method is used for the second drive.

After driving, the subjects are asked to complete a subjective evaluation questionnaire and a free-answer questionnaire about the system’s operability. The subjective evaluation questionnaire asks, “How did you feel about the operation of the electric wheelchair?”. Subjects are asked to answer the question on a five-point scale (Likert scale), as shown below. In the free-answer questionnaire, subjects are asked to describe the ride’s situation in detail.

1.Very difficult2.Difficult3.Neutral4.Easy5.Very easy

A Wilcoxon signed-rank test is performed on the operability of each group obtained via questionnaire evaluation at a significance level of 5%.

#### 4.2.2. Results

Figure 12 shows the results of the questionnaire and the Wilcoxon signed-rank test. The results of subjective evaluation by the conventional method were that one person scored 5, six people scored 4, one person scored 3, and six people scored 2. In contrast, 10 people scored 5 points, and 4 people scored 4 points for the proposed method. The Wilcoxon signed-rank test results show a statistically significant difference (*p* = 0.0002) in operability between the traditional and the proposed methods, confirming that subjects are more comfortable with the proposed method. Some of the results of the open-answer questionnaire are shown below:I felt that the second drive was more straightforward to operate than the first drive because the electric wheelchair turned with the timing I expected.The first drive was reassuring because there was a delay before the electric wheelchair started to turn at the corner, but I felt the delay was bothersome once I got used to the operation. I also made mistakes in the timing of turning. The second time, the electric wheelchair made the turn when I wanted to turn, so I operated the wheelchair without worrying about the wrong timing.In the first drive, the electric wheelchair frequently stopped when turning, but in the second drive, the wheelchair turned smoothly and on the intended path, so I drove with peace of mind.The first drive had delays when turning right and left, so it was difficult to anticipate these delays.The first drive sometimes got closer to the wall than expected at turns.

Next, the horizontal and vertical eye and head angles, the depth data measured by LRF, and the model outputs for the three subjects during electric wheelchair driving are shown in Figure 13, Figure 14 and Figure 15. In each figure, (a) is the traditional method using gaze dwell time, and (b) is the proposed method combining the gaze dwell time method and the visual intentions estimation model.

The output label means the control commands output by the control system. “Forward”, “Right”, and “Left” are output by the 1DCNN-LSTM model, and the “Delay” is output by the gaze dwell time method.

Each figure (a) shows that a time lag occurs from the time that the subject gazes in the direction of movement until the electric wheelchair runs. On the other hand, each figure (b) shows that the proposed method reduces the time lag compared to the traditional method. Especially in Figure 14a, subject B looked in a different direction from the one he had been looking at during the turning, which caused frequent delays due to the fixation detection using the gaze dwell time method. In contrast, Figure 14b shows a noticeable decrease in time lag occurrences. Subject B also looked away to the left while driving forward 6 s, 38 s, and 77 s after the start of driving in Figure 14b. However, Subject B did not move to the left because he turned his eyes back in the forward direction within the decision time required for the fixation detection by the gaze dwell time method.

In addition, subjects were instructed to read the poster displayed at a point about 20 s after they started running. When all subjects gazed at the poster, the electric wheelchair rotated in the direction they were gazing at, in the traditional method. In contrast, the electric wheelchair stopped on the spot without rotating in the proposed method.

However, in Figure 15b, the wheelchair stopped twice during the right turn between 7 s and 12 s. When the electric wheelchair stopped, subject C’s eye movement range was 0.288 rad to 0.476 rad on the X-axis and −0.006 rad to 0.155 rad on the Y-axis, and the depth data to the point where the eye/head was facing ranged from 1.56 m to 1.86 m. Subject A’s eye movement in the same period ranged from 0.234 rad to 0.710 rad on the X-axis and −0.037 rad to −0.110 rad on the Y-axis, with depth data ranging from 1.26 m to 5.60 m. Subject B’s eye movement ranged from 0.128 rad to 1.108 rad on the X-axis and 0.159 rad to 0.375 rad on the Y-axis, with depth data ranging from 1.09 m to 5.60 m.

The eye movement range and depth data values of subject C were smaller than those of the other subjects, indicating that subject C was gazing at the wall in front of him when the electric wheelchair stopped, and was not looking at an open space such as a hallway.

## 5. Discussion

In the model evaluation experiment described in Section 4.1, the 1DCNN-LSTM using the eye and head data sets classified visual intentions with the highest accuracy. Therefore, adding information on head movement as well as eye movement to the feature vector is presumed to be effective in estimating visual intentions.

In this study, the F1-score value improved with the length of the input time series, and each model’s F1-score reached its peak value within 2 s to 4 s. Festor et al. [67] also reported that visual intentions were determined with 65% accuracy with a 0.6 s input time series, and accuracy reached a peak of 92% with a 3.3 s input time series. These results indicate that estimating the visual intentions is possible by learning time series patterns from the natural eye and head behavior data. Furthermore, they suggest that a temporal component is essential for accurate estimation, and that the presence of characteristic eye and head movement patterns for estimating visual intentions may be included within 4 s.

The PFI results showed that the eye rotation angle vector is the most important feature contributing to the estimation of visual intentions. The next is the distance to the point where the eyes/heads are facing, suggesting that information about the external environment also contributes to the accurate estimation. The high ranking of depth information is considered due to the depth data during poster gazing and turning. In Figure 13b, Figure 14b and Figure 15b, the depth data to the point where the eyes and head were facing when gazing at the poster ranged from 0.49 m to 1.83 m for Subject A, 0.72 m to 1.89 m for Subject B, and 0.67 m to 1.46 m for Subject C. In contrast, the depth data during the left turn ranged from 1.47 m to 5.60 m (first left turn) and 1.23 m to 5.60 m (second left turn) for Subject A, 1.46 m to 5.60 m (first left turn) and 1.25 m to 5.60 m (second left turn) for Subject B, 1.63 m to 3.11 m (first left turn) and 1.37 m to 5.60 m (second left turn) for Subject C, suggesting that they were looking at the hallway. These results indicate that the depth data mainly show a difference between looking at the poster and turning a corner.

Next, the questionnaire results from the electric wheelchair running experiment showed that our implementation of the control system (1DCNN-LSTM model + gaze dwell time method) was more convenient to operate than the traditional method using only the gaze dwell time method. In the free-answer questionnaire, the subjects answered that the proposed control system allowed them to turn at the intended time and drive smoothly without getting too close to a wall when turning. These results were also reflected in the driving data recorded during the experiment. In the case of Subject B, he tended to move his eyes frequently to check his surroundings while turning. Thus, the system’s operability was decreased when using the traditional method because the estimation of visual intent was performed many times, and the turning operation was interrupted during the estimation. Hence, the control method based on gaze dwell time requires more effort to keep the gaze fixation [1]. On the other hand, incorporating the visual intentions estimation model into the traditional method enabled a real-time estimation, which reduced the delay time until the start of turning and minimized the occurrence of a time lag. Therefore, the system’s operability was improved because the subject could turn right/left, move forward, and stop at the timing intended, without the effort of keeping gaze fixation and assuming the delay time.

Moreover, the electric wheelchair stopped several times when turning a corner during operation with the proposed control system. An analysis of eye movements and depth data while driving revealed that the subject was gazing at a wall within 1.5 m to 2.0 m, and not in the hallway. Thus, we infer that the proposed model estimated that the subject was gazing at an object, and the stop command was output. Such a malfunction caused by the subject’s behavior seems to occur easily.

However, since the visual intentions estimation model uses only depth data obtained from the depth sensor as information on the external environment, the model cannot recognize the object seen by the user. Hence, we need to take an egocentric video and use object detection to solve that problem. For example, if the user looks at a wall while turning the electric wheelchair, the system does not stop the wheelchair, whereas if the user focuses on a specific object, the system stops the wheelchair. Therefore, the visual intentions estimation model is expected to adapt to various driving scenarios by using the information obtained from object detection.

Finally, at the current stage of our study, the gaze dwell time method was used to suppress the effects of eye movements on electric wheelchair driving that included observational intentions such as looking away and checking the surroundings, but these effects are insufficient. Although the proposed model outputs only four behavioral intentions (forward, right turn, left turn, and stop), the model can increase the number of tasks estimated through data collection and labeling for each driving scenario. Thus, we need to collect more data to estimate the observational intentions. Further research will be conducted to create data sets for various driving scenes, such as avoiding obstacles, checking left and right at crossroads, and human traffic, for the practical use of the electric wheelchair control system.

## Figures and Tables

**Figure 1 sensors-23-04028-f001:**
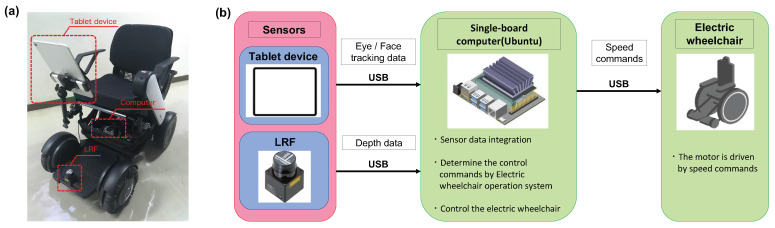
Structure of the eye-controlled electric wheelchair: (**a**) Appearance of the WHILL; (**b**) System components.

**Figure 2 sensors-23-04028-f002:**
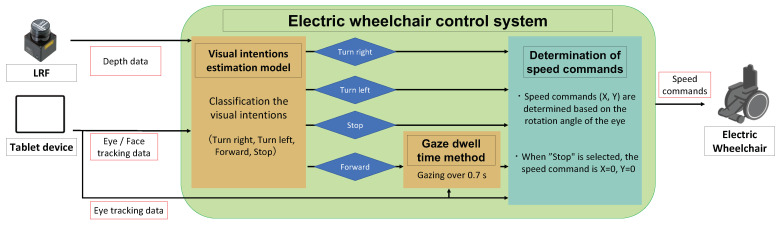
Block diagram of the electric wheelchair control system.

**Figure 3 sensors-23-04028-f003:**
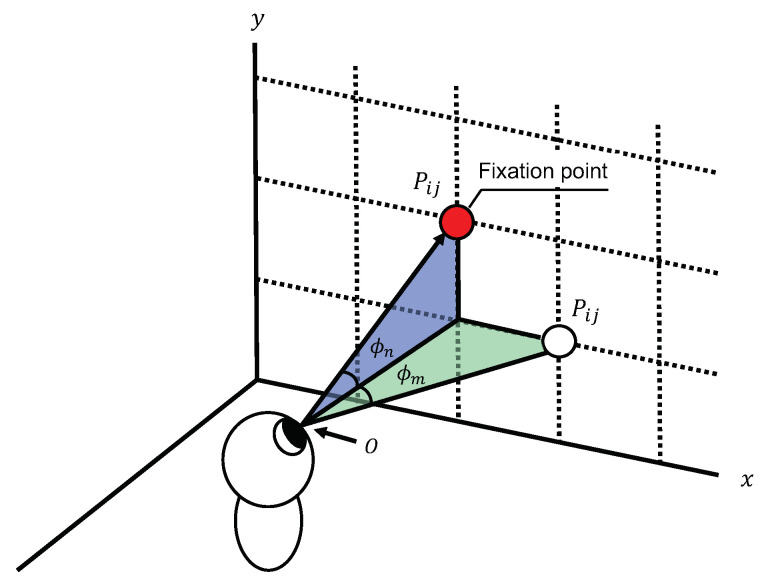
The rotation angles $\phi_{m}$ and $\phi_{n}$ between the fixation point and the other points.

**Figure 4 sensors-23-04028-f004:**
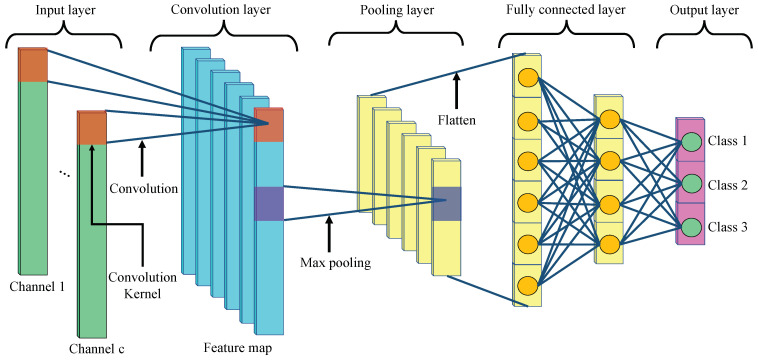
Architecture of the 1DCNN model.

**Figure 5 sensors-23-04028-f005:**
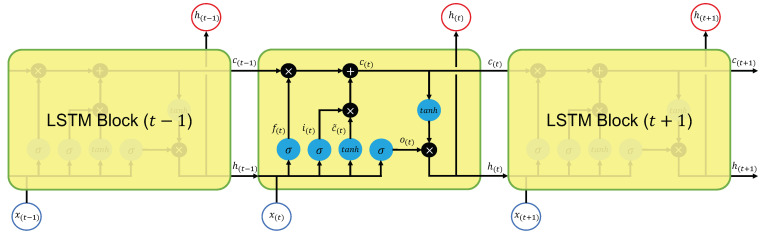
Architecture of the LSTM block.

**Figure 6 sensors-23-04028-f006:**
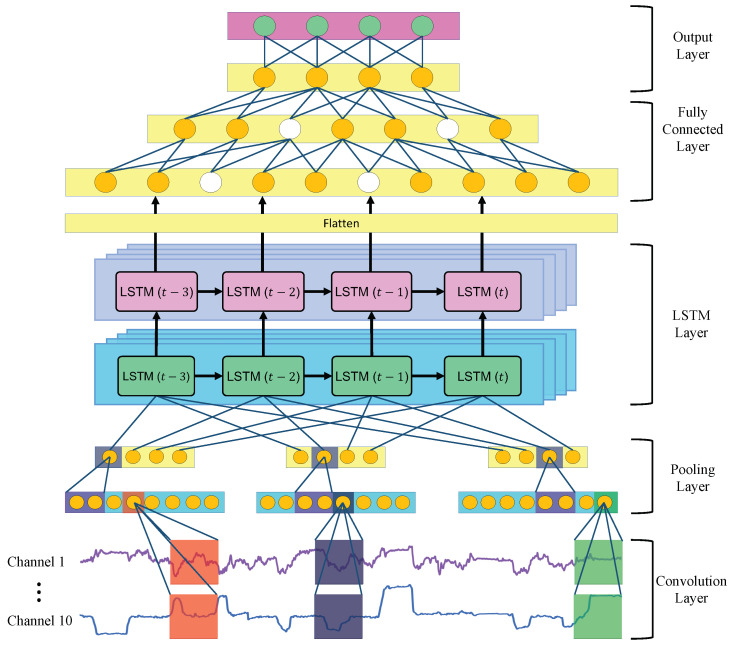
Architecture of the 1DCNN-LSTM.

**Figure 7 sensors-23-04028-f007:**
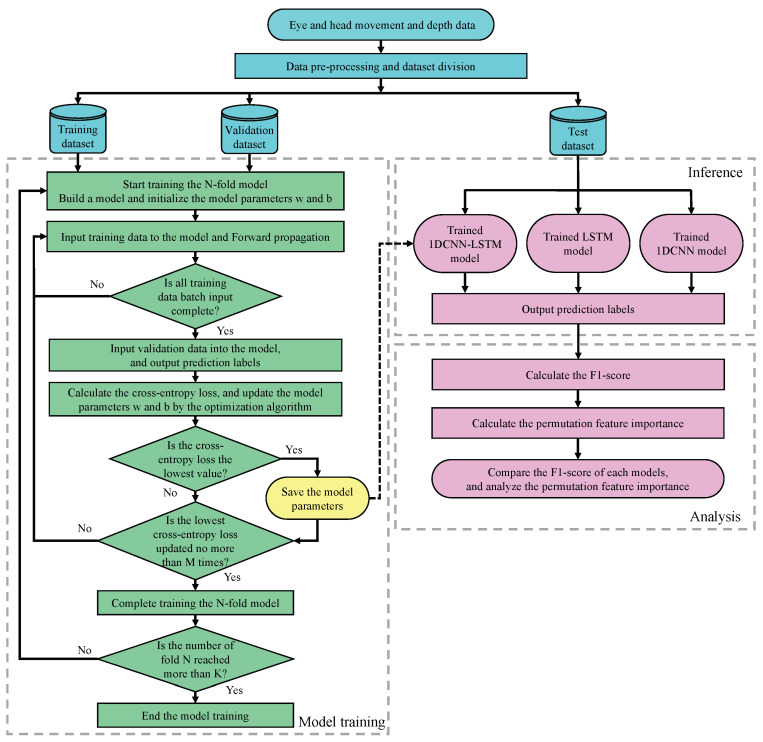
Flowchart of the visual intention estimation method.

**Figure 8 sensors-23-04028-f008:**
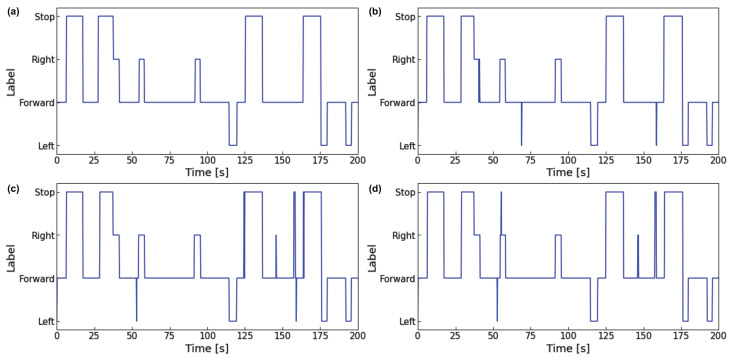
Comparison of output results of each model (0 s to 200 s): (**a**) Ground truth; (**b**) Output of 1DCNN-LSTM; (**c**) Output of 1DCNN; (**d**) Output of LSTM.

**Figure 9 sensors-23-04028-f009:**
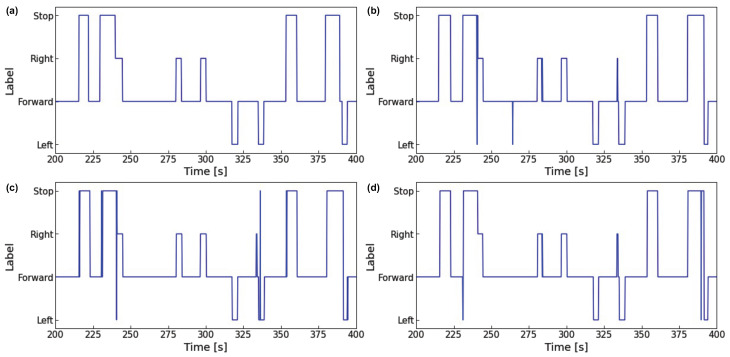
Comparison of output results of each model (200 s to 400 s): (**a**) Ground truth; (**b**) Output of 1DCNN-LSTM; (**c**) Output of 1DCNN; (**d**) Output of LSTM.

**Figure 10 sensors-23-04028-f010:**
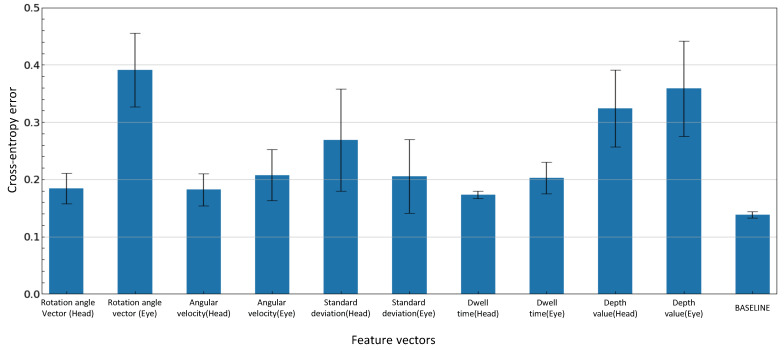
Result of permutation feature importance.

**Figure 11 sensors-23-04028-f011:**
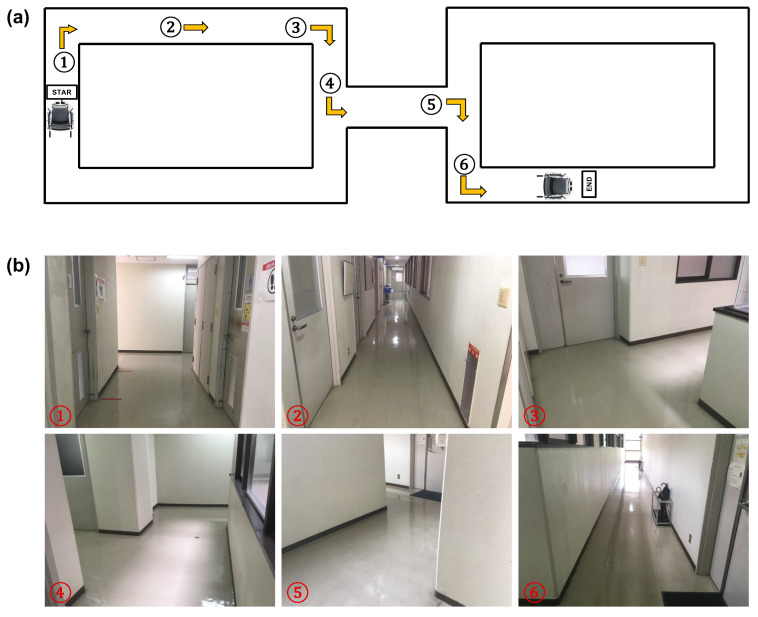
Route map of the experiment: (**a**) Overhead view of the route map of the experiment; (**b**) Actual view of the driving route. The numbers in the diagram indicate the actions to be taken by the subjects. For number 1 in the diagram, the subjects go straight and then turn right at the end of the street. For number 2, the subjects go straight ahead. When they pass a poster on the wall, they have to read it. For number 3, the subjects turn right at the corner. For number 4, the subjects go straight and then turn left at the first corner. For number 5, the subjects go straight and then turn right at the end of the street. For number 6, the subjects turn left at the corner and go straight to the finish line. Note: Each figure in the actual view of the driving route (**b**) corresponds to the number shown in the overhead view (**a**).

**Figure 12 sensors-23-04028-f012:**
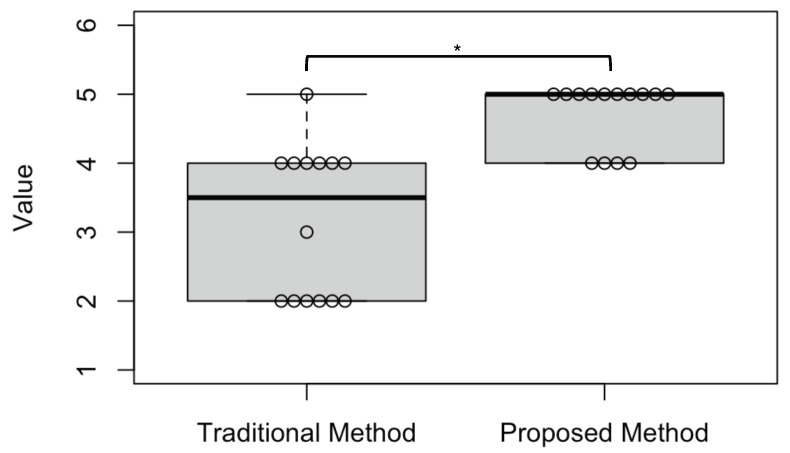
Box-and-whisker diagram of subjective evaluation. * The bold line in each box represents the median value of the score. The bottom side of the box represents the lower quartile, and the top side indicates the upper quartile. The whisker extending from the box indicates the maximum value that is not considered an outlier. The circles in each box indicate the number of subjects who scored points.

**Figure 13 sensors-23-04028-f013:**
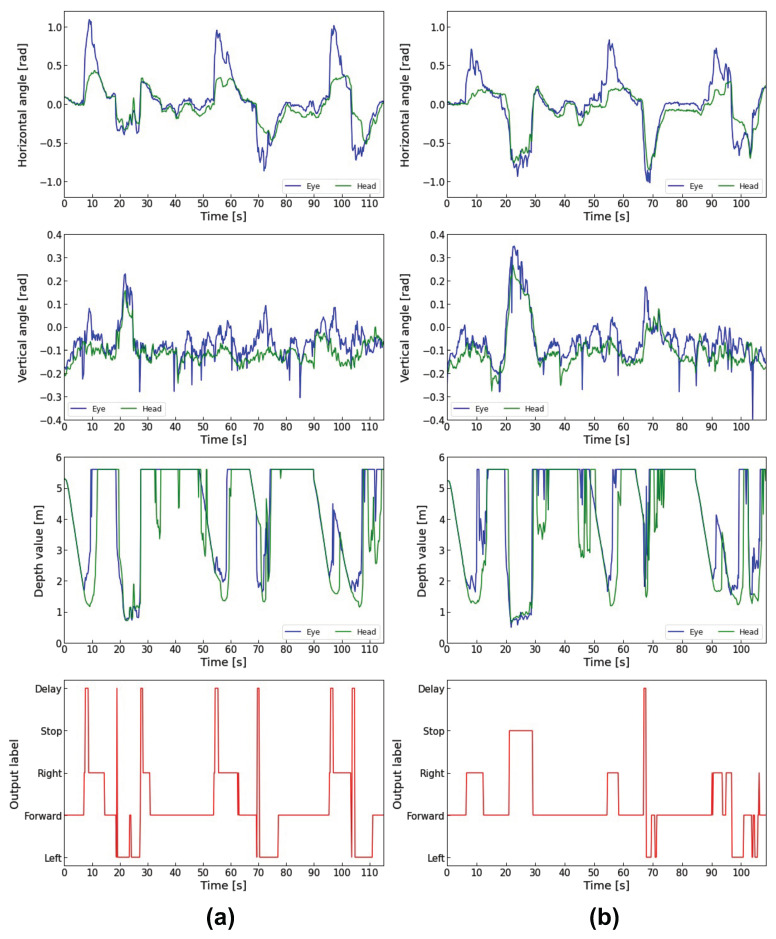
Comparison of Subject A’s driving data: (**a**) Driving data in the traditional method; (**b**) Driving data in the proposed method.

**Figure 14 sensors-23-04028-f014:**
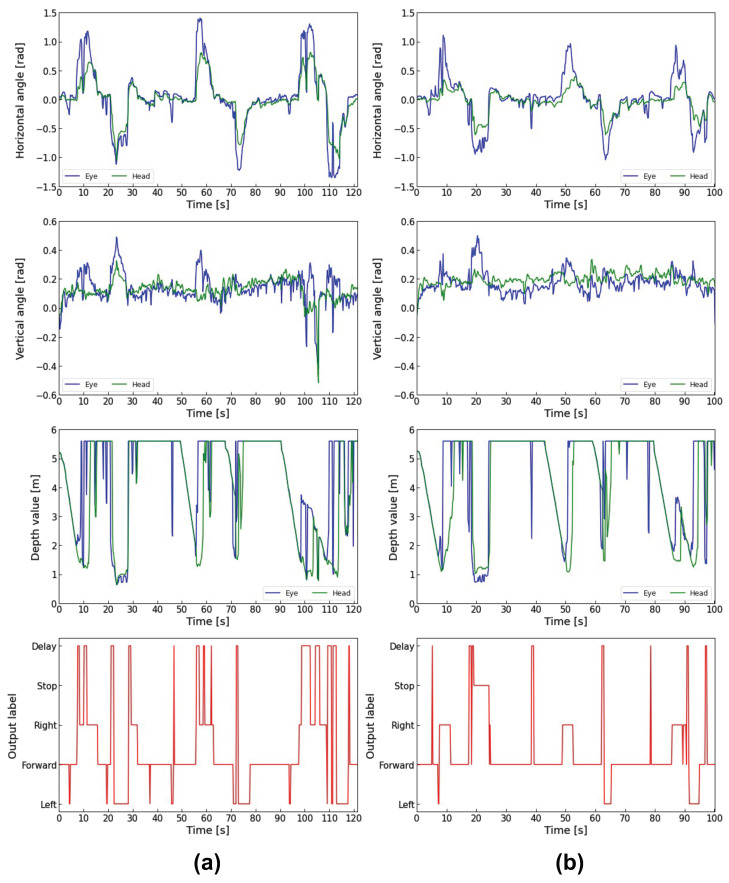
Comparison of Subject B’s driving data: (**a**) Driving data in the traditional method; (**b**) Driving data in the proposed method.

**Figure 15 sensors-23-04028-f015:**
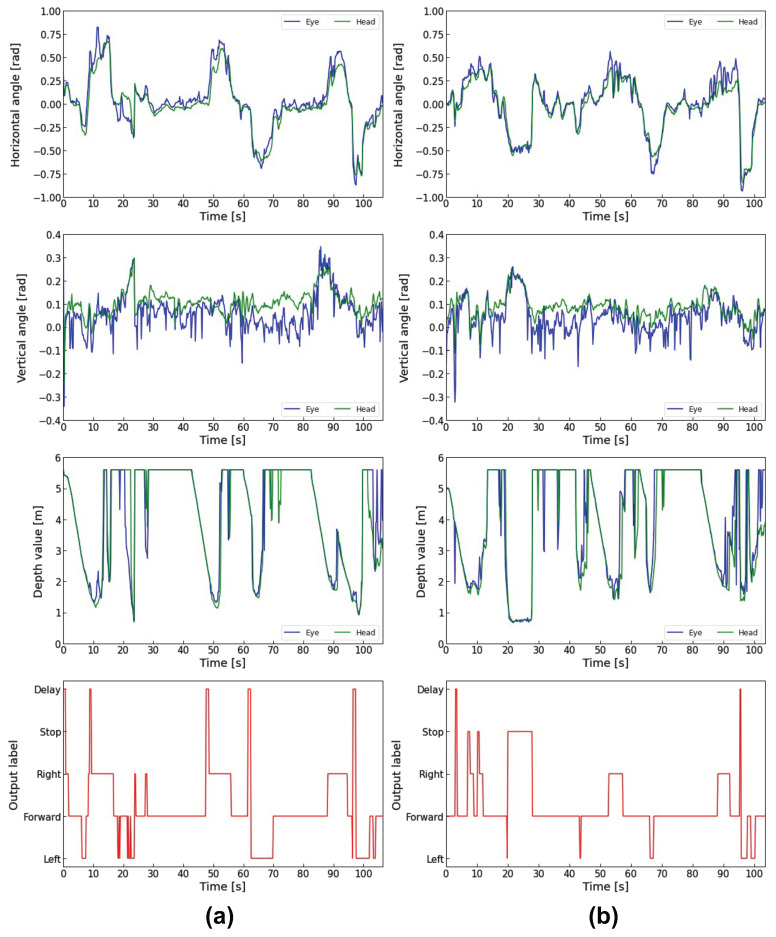
Comparison of Subject C’s driving data: (**a**) Driving data in the traditional method; (**b**) Driving data in the proposed method.

**Table 1 sensors-23-04028-t001:** Main differences in the structure of each model.

Models	Convolutional and Pooling Layers	Recurrent Neural Networks
1DCNN	∘	−
LSTM	−	∘
1DCNN-LSTM	∘	∘

**Table 2 sensors-23-04028-t002:** Comparison of the F1-score of each model.

Models (Data Sets)	Time Step				
	0.5 s	1 s	2 s	3 s	4 s
	Mean of *F*1*-Score* (Std)	Mean of *F*1*-Score* (Std)	Mean of *F*1*-Score* (Std)	Mean of *F*1*-Score* (Std)	Mean of *F*1*-Score* (Std)
1DCNN-LSTM (head + eye)	0.870 (0.012)	0.907 (0.003)	0.910 (0.004)	0.918 (0.003)	0.904 (0.007)
1DCNN-LSTM (head)	0.848 (0.010)	0.864 (0.013)	0.882 (0.008)	0.902 (0.003)	0.898 (0.006)
1DCNN-LSTM (eye)	0.878 (0.008)	0.895 (0.004)	0.895 (0.008)	0.907 (0.004)	0.902 (0.008)
LSTM (head + eye)	0.888 (0.007)	0.905 (0.010)	0.909 (0.004)	0.912 (0.004)	0.906 (0.011)
LSTM (head)	0.836 (0.011)	0.848 (0.007)	0.871 (0.010)	0.880 (0.011)	0.887 (0.010)
LSTM (eye)	0.865 (0.004)	0.897 (0.009)	0.894 (0.006)	0.899 (0.007)	0.900 (0.005)
1DCNN (head + eye)	0.875 (0.008)	0.904 (0.009)	0.909 (0.006)	0.915 (0.007)	0.903 (0.007)
1DCNN (head)	0.850 (0.005)	0.861 (0.016)	0.890 (0.006)	0.883 (0.009)	0.912 (0.009)
1DCNN (eye)	0.862 (0.012)	0.893 0.011)	0.906 (0.003)	0.903 (0.006)	0.903 (0.007)

**Table 3 sensors-23-04028-t003:** Raw data of the scores in 1DCNN-LSTM model (time step = 3 s, data sets = head + eye).

Number of Fold	Label Name	*Precision*	*Recall*	*F*1*-Score*	Mean of *F*1*-Score* (Std)	Number of Test Data
1	Left	0.893	0.896	0.895	0.921 (0.036)	327
	Forward	0.981	0.970	0.975		3250
	Right	0.922	0.846	0.882		292
	Stop	0.890	0.975	0.931		642
2	Left	0.884	0.911	0.898	0.921 (0.034)	327
	Forward	0.980	0.967	0.974		3250
	Right	0.909	0.860	0.884		292
	Stop	0.893	0.963	0.927		642
3	Left	0.890	0.890	0.890	0.913 (0.040)	327
	Forward	0.983	0.962	0.972		3250
	Right	0.861	0.870	0.865		292
	Stop	0.883	0.972	0.925		642
4	Left	0.906	0.884	0.895	0.919 (0.037)	327
	Forward	0.977	0.971	0.974		3250
	Right	0.882	0.870	0.876		292
	Stop	0.908	0.953	0.930		642
5	Left	0.918	0.856	0.886	0.915 (0.041)	327
	Forward	0.975	0.972	0.973		3250
	Right	0.886	0.849	0.867		292
	Stop	0.904	0.964	0.933		642

## Data Availability

The data sets presented in this study are available on request from the corresponding author. The data sets are not publicly available due to data privacy. The code used in the inference accuracy comparison experiments is available in the following repositories. https://github.com/sabo0202/Visual-intentions-estimation-model (accessed on 10 April 2023).

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
