# Peer review of "Intelligent Eye-Controlled Electric Wheelchair Based on Estimating Visual Intentions Using One-Dimensional Convolutional Neural Network and Long Short-Term Memory"

_sensors, 2023, doi:10.3390/s23084028_

Round 1
Author Response
Response to Reviewer 1 Comments
Point1: A The age range of subjects tested in the effectiveness evaluation of 1DCNN-LSTM in 4.1 can increase middle age and old age; To test the stability of the system, while making the test results more convincing.
Response 1: Due to the fact that additional experiments require a certain amount of subjects and time, we are very sorry to have to forego the points raised due to the revision period. However, the participation of middle-aged and elderly subjects in the experiments you mentioned is an important factor for the use of electric wheelchairs and for future research. Therefore, we would like to include the points raised in our next paper.
The authors would like to thank the reviewers for their time and effort in reviewing this manuscript.
Reviewer 2 Report
In this study, authors investigate intelligent eye-controlled electric wheelchairs using 1DCNN-LSTM. The authors conducted a topological characteristic of the deep learning model. However, several flaws should be revised before publication.
A flowchart or tables representing the process (codes, comparisons, analysis) should be included in detail to improve the reader's comprehension.
Authors should supplement more words in their study and the detailed parameters in the process of data analysis.
Please provide more details about the design scheme method and tell readers the difference and advantages of this model with other approaches.
You can carry out a permutation test (Fisher exact test) that requires no distribution assumptions (require enough computational power). For multiple testing corrections, Benjamini-Hochberg is a widely used FDR algorithm with the best FP and FN controlling performance under independence assumptions.
For reproducibility, the author should release the code in the cloud or other repositories, such as GitHub, so that users can repeat it.
The quality of the figures is not good and needs to be modified.
There are a few minor language issues that need to be addressed.
Author Response
Response to Reviewer 2 Comments
Point 1: A flowchart or tables representing the process (codes, comparisons, analysis) should be included in detail to improve the reader's comprehension.
Response 1: A flowchart representing the training and inference of the models proposed in manuscript and the comparison and analysis of accuracy has been added to Section 3.6 (page 29) of Materials and Methods.
Point 2: Authors should supplement more words in their study and the detailed parameters in the process of data analysis.
Response 2: In Section 4.1.4, a table of the raw data that led to the calculation of the average F1 score for the 1DCNN-LSTM model with the best accuracy (time step = 3 s, data sets = head + eye) and a description of the results has been added. A detailed explanation of the terms "head + eye", "head" and "eye" in Table 2 has also been added to the text on page 16, lines 1-8.
Point 3: Please provide more details about the design scheme method and tell readers the difference and advantages of this model with other approaches.
Response 3: A description of 1DCNN-LSTM and how it differs from other models has been added on page 10, lines 29 to 11, line 6, and Table 1.
Point 4: You can carry out a permutation test (Fisher exact test) that requires no distribution assumptions (require enough computational power). For multiple testing corrections, Benjamini-Hochberg is a widely used FDR algorithm with the best FP and FN controlling performance under independence assumptions.
Response 4: We are grateful for your suggestions with regard to the test method.
The subjective rating questionnaire in the operability evaluation experiment in the paper was conducted on a five-point ordinal scale. In the experiment, the 14 subjects always performed two electric wheelchair rides; the first time they operated the electric wheelchair using the traditional method and the second time they used the proposed method.
As the operability rating was on a five-point ordinal scale and the subject groups were matched groups, a statistical test was performed using the Wilcoxon signed-rank test, a non-parametric test that does not require normality, to test whether there was a significant difference between the two groups.
Point 5: For reproducibility, the author should release the code in the cloud or other repositories, such as GitHub, so that users can repeat it.
Response 5: We have uploaded to the corresponding author's (Sho Higa) Github repository the code used in the model inference accuracy comparison experiments conducted in Section 4.1. The URL is shown below.
https://github.com/sabo0202/Visual-intentions-estimation-model
Point 6: The quality of the figures is not good and needs to be modified.
Response 6: The resolution of Figure 12 was low and blurred, so the resolution has been increased.
Point 7: There are a few minor language issues that need to be addressed.
Response 7: In order to meet the deadline, corrections relating to other points raised have been prioritised. We sincerely apologise, but if there are sentences that contain clear problems, we would appreciate it if you could point out the relevant sections.
The authors would like to thank the reviewers for their time and effort in reviewing this manuscript.
Reviewer 3 Report
The problems of incorrect recognition in gaze motion in operated this way electric wheelchair are in focus of the study. The deep learning model used visual time series patterns from eye and head movement data is proposed, and also an electric wheelchair control system is developed and tested in experiment. The proposed model show improved behavior compared to other models.
The problem is clearly introduced; the material is presented quite clearly. Achievements that are able to attract readers are visible.
I can say that the paper is written well, if you do not pay attention to unexplained abbreviations. They spoil the perception and must be decrypted at the first application.
1. 1DCNN-LSTM. In the title, it is better to replace it with a verbal synonym or with full version (one-dimensional convolutional neural network), and then introduce an abbreviation in the text, providing an explanation: what does it mean.
2. EEG signals
Author Response
Point 1: I can say that the paper is written well, if you do not pay attention to unexplained abbreviations. They spoil the perception and must be decrypted at the first application.
1. 1DCNN-LSTM. In the title, it is better to replace it with a verbal synonym or with full version (one-dimensional convolutional neural network), and then introduce an abbreviation in the text, providing an explanation: what does it mean.
2. EEG signals
Response 1: The authors would like to thank the reviewers for their time and effort in reviewing this manuscript.
As you suggested, the title has been revised to include the official name of 1DCNN-LSTM. Accordingly, an explanation of 1DCNN-LSTM has been added from page 10, line 29 to page 11, line 6 in the text.
In addition, the official name of EEG (electroencephalography) has been added to the text on page 1, line 7.